# Impact of Multi-Walled CNT Incorporation on Dielectric Properties of PVDF-BaTiO$_3$ Nanocomposites and Their Energy Harvesting Possibilities

**Abu Sadat M. Iftekhar Uddin [1], Dongin Lee [2], Chanseob Cho [3,*] and Bonghwan Kim [1,*]**

[1] School of Electronic and Electrical Engineering, Daegu Catholic University, Gyeongsan 38430, Gyeongbuk, Korea; iftekhar@metrouni.edu.bd
[2] Department of Information and Communication Engineering, Yeungnam University, Gyeongsan 38541, Gyeogbuk, Korea; dilee65@yu.ac.kr
[3] School of Electronic Engineering, Kyungpook National University, Daegu 41566, Daegu, Korea
* Correspondence: chocs@knu.ac.kr (C.C.); bhkim@cu.ac.kr (B.K.)

**Abstract:** The current study investigated the fabrication of multi-walled carbon nanotubes (MWCNTs) adhering to Barium titanate (BaTiO$_3$) nanoparticles and poly(vinylidene fluoride) (PVDF) nanocomposites, as well as the impact of MWCNT on the PVDF-BaTiO$_3$ matrix in terms of dielectric constant and dielectric loss with a view to develop a high performance piezoelectric energy harvester in future. The capacity and potential of as-prepared nanocomposite films for the fabrication of high-performance flexible piezoelectric nanogenerator (PNG) were also investigated in this work. In particular, five distinct types of nanocomposites and films were synthesized: PB (bare PVDF–BaTiO$_3$), PBC-1 (PVDF–BaTiO$_3$-0.1 wt% CNT), PBC-2 (PVDF–BaTiO$_3$-0.3 wt% CNT), PBC-3 (PVDF–BaTiO$_3$-0.5 wt% CNT), and PBC-4 (PVDF–BaTiO$_3$-1 wt% CNT). The dielectric constant and dielectric loss increased as MWCNT concentration increased. Sample PBC-3 had the optimum dielectric characteristics of all the as-prepared samples, with the maximum output voltage and current of 4.4 V and 0.66 μA, respectively, with an applied force of ~2N. Fine-tuning the BaTiO$_3$ content and thickness of the PNGs is likely to increase the harvester's performance even more. It is anticipated that the work would make it easier to fabricate high-performance piezoelectric films and would be a suitable choice for creating high-performance PNG.

**Keywords:** PVDF–BaTiO$_3$-xCNT nanocomposite; piezoelectric nanogenerator; flexible; dielectric properties

## 1. Introduction

The fast progression to the semiconductor industry and microelectronics have led to functional devices, including implantable medical appliances, personal wearable electronics, or the Internet of Things (IoT), to be at every point in every corner, playing a major part in everyday life [1–3]. However, determining how to power them has proven to be a significant difficulty, as typical powering methods rely on batteries, which have limited capacity and longevity. As a consequence, piezoelectric materials have received great attention because of their potential as a portable power source, and the use of a piezoelectric nano-generator (PENG) is considered to be one of the most promising solutions because it can convert the ubiquitous and versatile mechanical energy into electricity [4–12]. Among various organic flexible piezoelectric materials (such as nylon, polylactic acid (PLA), polylactic glycolic acid (PLGA)), poly(vinylidene fluoride) (PVDF) and its copolymer (for example, trifuoroethylene (TrFE), hexafluoropropylene (HPF), or chlorotrifluoroethylene (CTFE)) with their remarkable mechanical flexibility, bio-compatibility, transparency, chemical stability and high breakdown strength, are regarded to be the most important candidates [13–17].

However, despite having a high piezoelectric voltage constant ($g_{33}$), PVDF has a low dielectric constant ($d_{33}$), whereas both the $g_{33}$ and $d_{33}$ parameters are critical for extracting the maximum energy from PVDF [18].

Researchers have taken a variety of approaches to increase dielectric constant and decrease dielectric loss in order to obtain nanocomposites with excellent dielectric properties and develop high performance piezoelectric nanogenerators (PNGs). To endow polymer nanocomposites with high dielectric constant, researchers have homogeneously dispersed ferroelectric ceramic particles or fibers with high dielectric constant [19–23] and/or incorporated conductive fillers such as metal particles, carbon-based materials, and semiconductor, into the polymer matrix [24–30]. Among various ceramic materials including $PbZr_{1-x}Ti_xO_3$ (PZT), PZT:La (PLZT), $PbTiO_3$ (PT), and $Pb(Mg_{1/3}Nb_{2/3})O_3$ (PMN), lead-free barium titanate ($BaTiO_3$) is the most environmentally friendly of the ceramic materials, and it also has high piezoelectric and ferroelectric characteristics, as well as a high dielectric constant [19,23]. Nanocomposites of piezoelectric polymers and ceramic can improve the PNG's energy-scavenging properties by providing relatively high $g_{33}$ and $d_{33}$ values, as well as high flexibility; however, a lower amount of ceramic filer can reduce the nanocomposite's energy harvesting properties, as the positive and negative piezoelectric co-efficient of ceramic and polymer can cancel each other's effect, reducing the nanocomposite's energy harvesting properties. [20]. To attain optimal dielectric characteristics, ceramic filler content is normally required to be no less than 50 vol% [31,32]. Percolative nanocomposites, on the other hand, excel due to their high dielectric constant obtained at very low filler content while retaining relatively high mechanical strength. Multi-walled carbon nanotubes (MWCNT) are more preferred as dielectric fillers in achieving higher dielectric constant of nanocomposites due to their larger aspect ratio and higher electrical conductivity when compared with spherical and flake-shaped fillers. At low filler content, polymer filled with MWCNTs has been shown to have a high dielectric constant and mechanical strength [33].

The fabrication of multi-walled carbon nanotubes (MWCNTs) adhering to barium titanate ($BaTiO_3$) nanoparticles and poly(vinylidene fluoride) (PVDF) nanocomposites, as well as the impact of MWCNT on the PVDF-$BaTiO_3$ matrix in terms of dielectric constant and dielectric loss, were investigated in the current study. This study also looked at the capability and potential of as-prepared PVDF-$BaTiO_3$ and PVDF-$BaTiO_3$ with different concentrations of MWCNT nanocomposites for the construction of high-performance flexible piezoelectric nanogenerator (PNG). The incorporation of the optimal concentration of conductive filler into the PVDF-$BaTiO_3$ matrix was discovered to have substantial and intriguing effects on the structural and dielectric characteristics of the nanocomposite system. The conductive MWCNT filler not only aligned the dipoles in the PVDF matrix, but it also improved the nanocomposite's dielectric characteristics and energy harvesting properties. The as-prepared piezoelectric film is predicted to be a suitable choice for creating high-performance PNG and diverse applications in electronics devices.

## 2. Experimental Section

### 2.1. Materials

Multi-walled carbon nanotube (MWCNT) powder (D × L 110–170 nm × 5–9 μm) and N-N dimethylformamide (DMF, anhydrous 99.8%) solvent were purchased from Sigma-Aldrich Co. Inc. Barium titanate ($BaTiO_3$) powder was purchased from DaeJung Chemicals & Metals Co. Ltd. Poly(vinylidene fluoride) (PVDF) powder was purchased from Alfa Aesar. All the chemicals used in the synthesis process were of analytical grade and were used without further purification.

### 2.2. Preparation of PVDF–BaTiO$_3$-xCNT Solution

Three different solutions were prepared simultaneously in the following manner: 1.5 g PVDF powder was dispersed into 15 mL DMF solution under vigorous magnetic stirring for 5 h at a temperature of 55 °C; 0.83 g $BaTiO_3$ was dissolved into 5 mL DMF solution and

stirred at the same temperature of 55 °C and ultrasonicated for 5 min; 0.0015 g MWCNT was dissolved into 5 mL DMF solution at room temperature and ultrasonicated for 10 min. $BaTiO_3$ filler was kept 55 wt% with respect to the PVDF matrix in all the cases. After 5 h, the $BaTiO_3$ solution was mixed dropwise into the PVDF solution. Subsequently, MWCNT solution was added dropwise into the PVDF–$BaTiO_3$ solution to obtain the PVDF–$BaTiO_3$-0.1 wt% CNT composite solution. Afterward, varied amount of MWCNTs (with respect to PVDF content) were dispersed into PVDF–$BaTiO_3$ solution to obtain the 0.3, 0.5, and 1 wt% PVDF–$BaTiO_3$-xCNT nanocomposite solutions. The as-prepared solutions were further stirred at 700 rpm for 3 h and ultrasonicated for 5 min after every 1 h. Finally, the solutions were kept in vacuum chamber for 30 min to remove the bubbles from the solution. Bare PVDF–$BaTiO_3$ nanocomposite solution was also prepared following the similar process. The weight fraction of CNT was determined using the following equation

$$wt(\%) = \frac{w_c}{w_c + w_p} \times 100 \tag{1}$$

where $w_c$ and $w_p$ are the weights of the MWCNT and PVDF, respectively.

### 2.3. Synthesis of PVDF–$BaTiO_3$-xCNT Thin Films

To prepare the bare PVDF–$BaTiO_3$ and PVDF–$BaTiO_3$-xCNT thin films, fenced and cleaned Si substrate was placed on hot plate at 75 °C; importantly, the hotplate was carefully kept as flat as possible to obtain a uniform film. Then, the as-prepared solution was drop casted using doctor blade and dried for 3 h. Afterward, the dried nanocomposite film was carefully peeled off from the Si substrate. The thickness of the films was measured to be nearly ~50 μm. For the ease of working the as-prepared films were named as follows: PB (bare PVDF–$BaTiO_3$), PBC-1 (PVDF–$BaTiO_3$-0.1 wt% CNT), PBC-2 (PVDF–$BaTiO_3$-0.3 wt% CNT), PBC-3 (PVDF–$BaTiO_3$-0.5 wt% CNT), and PBC-4 (PVDF–$BaTiO_3$-1 wt% CNT).

### 2.4. Fabrication of PNGs

To fabricate the PNGs, Al (thickness: ~200 nm) as an electrode was deposited on both sides of the films using RF magnetron sputter system. The sample was then sliced into 3 cm × 1 cm (length × width) pieces, Cu wires were attached to the Al surface using silver (Ag) paste, and the sample slices were wrapped using two PET sheets. The overall synthesis procedure, schematic diagram, and optical image of the PNG are depicted in Figure 1.

### 2.5. Characterizations

Phase transition analysis was carried out with a high temperature X-ray diffraction (HT-XRD, Bruker AXS D8-Discover; Bruker, Berlin, Germany) system with Cu Kα (λ = 0.154056 nm) radiation and a 2θ scanning range of 10–80°. Fourier-transform infrared spectroscopy (FTIR) analysis was recorded using a Varian 2000 Scimitar spectrometer (Varian Inc., Palo Alto, CA, USA) in the range of 500–4000 $cm^{-1}$. The surface morphology of the as-prepared thin films and elemental analysis were examined via field emission scanning electron microscopy (FESEM, HITACHI S-4800; Hitachi, Tokyo, Japan) and electron dispersive spectroscopy (EDAX, HORIBA EX-250; Horiba, Tokyo, Japan). The electrical outputs and open-circuit voltage properties were measured using a functional oscilloscope (Tektronix TDS 2001C; Tektronix Inc., Beaverton, OR, USA). A radio frequency (RF) magnetron sputtering system (KVS C4055; Korea Vacuum Tech., Gimpo, Korea) was used to deposit aluminum on the nanocomposite films as bottom and top contacts. The dielectric properties of the film were measured using an impedance analyzer (KEYSIGHT E4990A; Keysight Technologies, Colorado Springs, CO, USA).

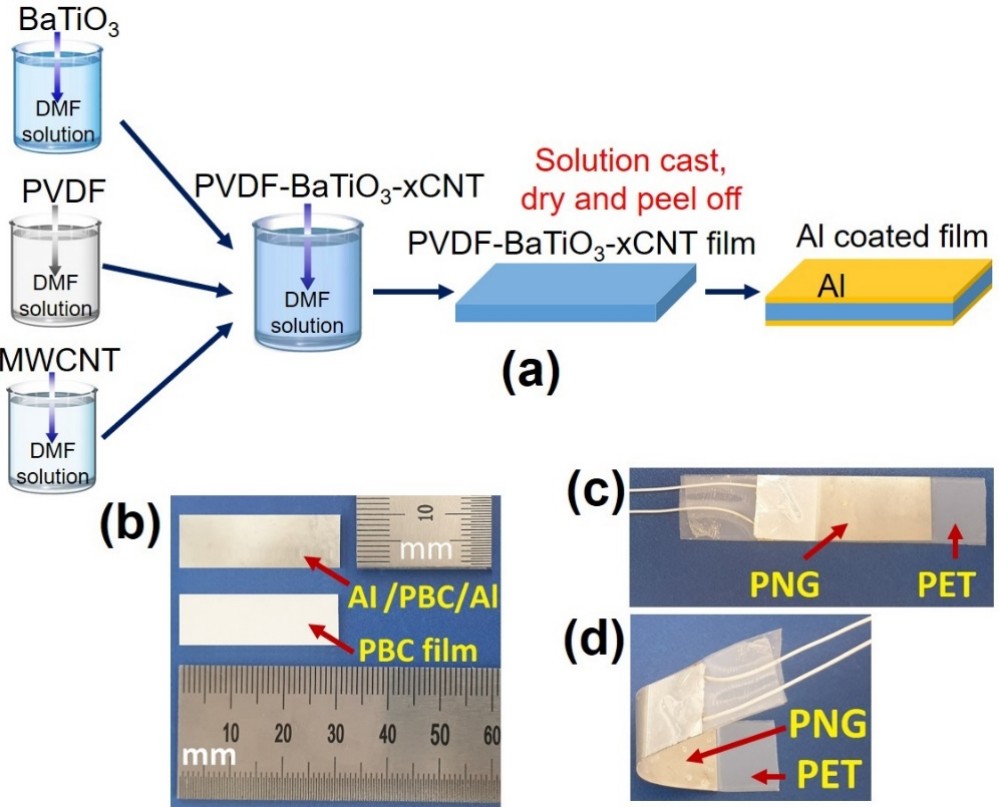

**Figure 1.** (**a**) Process flow and schematic view of the PVDF–BaTiO$_3$-xCNT (PBC) film and piezoelectric nanogenerator (PNG); (**b**) photograph of the as-prepared PBC film without and with Al coating. Film dimension is 3 × 1 cm square; (**c**) photograph of PNG after packaging with PET film; (**d**) photograph of the PNG device with good flexibility.

## 3. Results and Discussion

### 3.1. Morphology and Crystal Structure Analysis

The surface and cross-sectional morphologies of the as-synthesized piezoelectric films were characterized using FESEM, and the characterized output is shown in Figures 2 and 3. Brighter domains in the FESEM image in Figure 2a reflects the presence of heavier elements, specifically BaTiO$_3$ particles (~250 nm of mean diameter) that are both partially separated and partially agglomerated. All five as-prepared samples were analyzed and similar micrographs found. An enlarged view of PVDF surface is depicted in Figure 2b. In order to achieve high piezoelectric and dielectric properties, it is highly desirable to have an optimistic nanoscale level dispersion of MWCNT in the PVDF-BaTiO$_3$ matrix. However, MWCNTs are not apparent in any of the samples (PBC-1 to PBC-4), which could be owing to a low MWCNT content.

EDAX microanalysis (EDS mapping only) was performed on the bright domains in which the presence of barium (Ba), oxygen (O), and titanium (Ti) is confirmed. In fact, the existence of Ba, Ti, fluorine (F), O, and carbon (C) as shown in Figure 2c indicates the formation of high-purity nanocomposite material.

The cross-section FESEM view and EDAX microanalysis (including EDS mapping) of the sample PB after coating Al (both side as electrode) is shown in Figure 3. The average thickness of the film is measured to be nearly 60 μm (Figure 3a). Existence of Ba, Ti, F, O, C including Al is further confirmed and the elemental analysis is shown in Figure 3b–h.

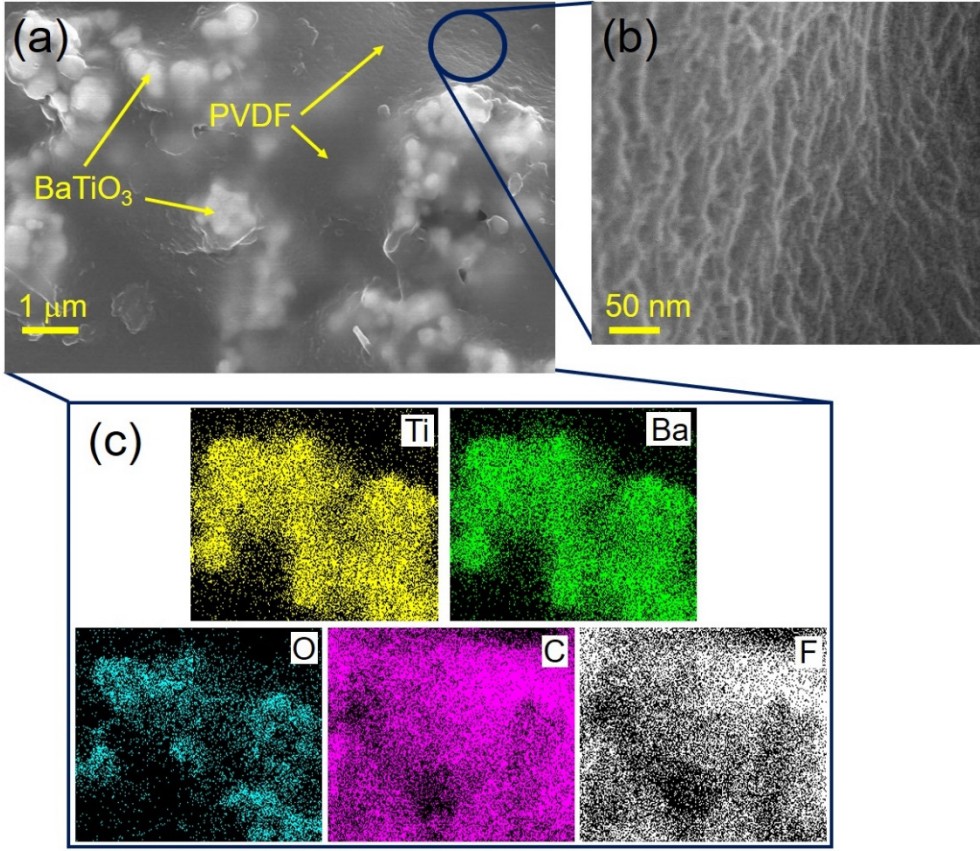

**Figure 2.** (**a**) FESEM micrograph of the as-prepared PBC film; (**b**) enlarged view of PVDF surface; (**c**) corresponding elemental mapping of the PBC film.

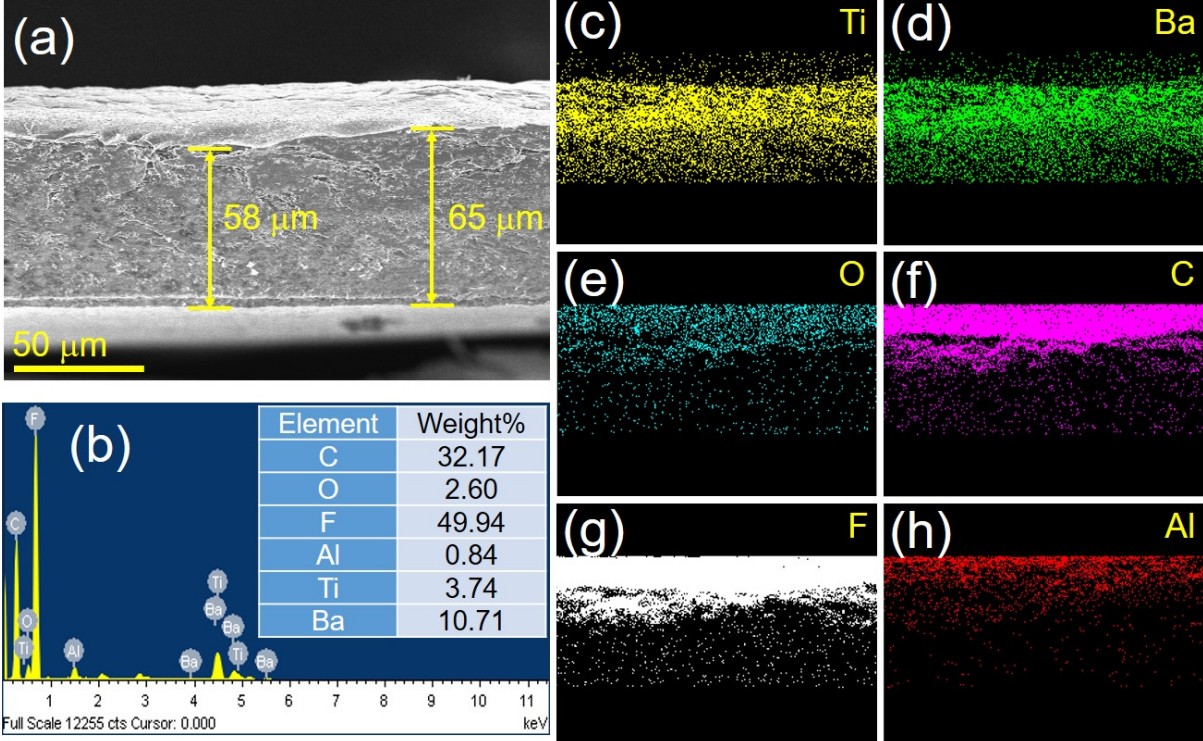

**Figure 3.** (**a**) Cross-section FESEM micrograph of the as-prepared PBC film; corresponding (**b**) EDS analysis; (**c–h**) elemental mapping of the PBC film.

The crystal structure and phase of the as-synthesized piezoelectric thin films were investigated using XRD analysis. The XRD pattern of each nanocomposite material is depicted in Figure 4. PVDF is a common piezoelectric material with good mechanical and piezoelectric characteristics. However, only the β-phase and γ-phase demonstrate piezoelectric behavior as a polycrystalline material [34]. The features XRD peak located at 20.1° corresponds to the β-phase of the PVDF. There was no α-phase detected in the PB film, which may be attributed to the conversion of α-phase to β-phase caused by the addition of BaTiO$_3$ content. According to some authors, the peak at 20.1° belongs to both the α- and β-phases, more specifically the (110) reflection of the α-phase and the (200)/(110) reflections of the β-phase [35]. When β-phase coexists with α-phase, the diffraction peak at 20.1° moves to a lower angle (for example, nearly at 19.9°), and a new diffraction peak may forms nearly at 20.7°, which corresponds to the to the (200)/(110) reflections of the β-phase [36]. However, no similar peaks were seen in any of the samples. A relatively small increase in that peak with increased MWCNT concentration as filler, on the other hand, suggests a direct impact on the PVDF polymorphism. The C-groups in the PVDF may have interacted with the fluorine (F) groups, causing them to align to one side and create a β configuration. The electroactive β phase of the nanocomposites corresponds to the planer all-trans (TTTT) conformation and can preferentially influence the dipole moments associated with the C-H and C-F bonds to be aligned in the direction perpendicular to the carbon backbone to give higher dipole moments per unit cell. On the other hand, the surface charges due to the presence of π-electrons including oxygen functionalized groups in MWCNT might interact with the CH$_2$ dipoles of the PVDF polymer chain and slightly increases the rate of nucleation of the electroactive β phase. The existence of the diffraction peaks (001), (101), (111), (200), (102), (112), (220), and (103) at 22.12°, 31.54°, 38.90°, 45.36°, 50.92°, 56.28°, 65.98°, and 75.04°, respectively, can be attributed to the BaTiO$_3$ tetragonal phase (JCPDS 05-0626). Splitting of the (200) peak into (200) and (002) was observed in all the samples, which may have disrupted the unit cell of the BaTiO$_3$ due to the presence of functional groups of MWCNT.

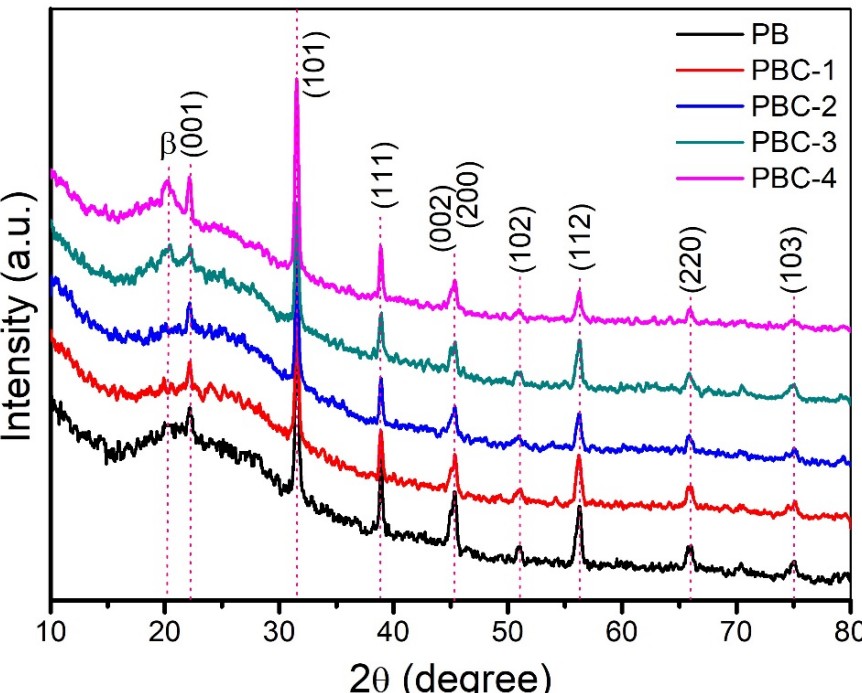

**Figure 4.** Representative XRD patterns of the as-prepared PVDF–BaTiO$_3$ (PB) and PVDF–BaTiO$_3$-xCNT (PBC) films.

The as-prepared films were further investigated using FTIR spectroscopy at room temperature (25 °C) in the wavenumber range of 500 cm$^{-1}$ to 4000 cm$^{-1}$, and the results are shown in Figure 5. Figure 5 was divided into Figure 5a (range: 500 cm$^{-1}$ to 1500 cm$^{-1}$) and Figure 5b (range: 1500 cm$^{-1}$ to 3500 cm$^{-1}$) for maximal clarity. The FTIR results are well matched with the XRD data and shed further light on the influence of MWCNT inclusion in the PVDF-BaTiO$_3$ matrix. The peaks of BaTiO$_3$ in the FTIR spectrum at 540 cm$^{-1}$, 600 cm$^{-1}$, 1094 cm$^{-1}$, 1340 cm$^{-1}$, and 1642 cm$^{-1}$ reveal Ti-O stretching, Ti-OH, COO-, OH, and Ba-OH bonds [37]. The 766 cm$^{-1}$ and 1234 cm$^{-1}$ characteristic bands correspond to the α-phase and γ-phase, respectively [38]. All of the samples had significant vibration peaks at 836 cm$^{-1}$, 875 cm$^{-1}$, and 1172 cm$^{-1}$, which were typical β-phase peaks [38]. With increasing MWCNT concentration, there was a little decrease in the intensity of the nonpolar α-phase and a slight rise in the β-phase, demonstrating the β-induction action inside the PVDF matrix and the stability of the electroactive polar β-phase as dominant [39,40]. The proportional rise in the β-phase with increased MWCNT loading might be attributed to MWCNT's strong effective ability for electroactive β-phase nucleation [40]. Carbonyl (C=O), carbon double bond (C=C), and alcohol (C-O) are indicated by stretching peaks at 1741 cm$^{-1}$, 1623 cm$^{-1}$, and 1162 cm$^{-1}$. This finding confirms the hexagonal structure of the CNTs [41]. The peak at 1623 cm$^{-1}$ can be associated with the stretching of the carbon nanotube backbone. It is usually assumed that these groups are located at defect sites on the sidewall surface [42,43]. The symmetric and asymmetric stretching of CH$_2$ are represented by the bands at 3038 cm$^{-1}$ and 2981 cm$^{-1}$, respectively [43]. The typical N-H stretching at 1400 cm$^{-1}$ shows that the DMF contamination may still be present in the film. Using FTIR spectrum, the β-phase content was calculated using the following Equation (2), and the results are shown in Table 1.

$$F(\beta) = \frac{A_\beta}{\frac{K_\beta}{K_\alpha} \times A_\alpha + A_\beta} \tag{2}$$

where $F(\beta)$ represents the β-phase content, $A_\alpha$ and $A_\beta$ are the absorption intensity, $K_\alpha$ and $K_\beta$ (6.1 × 10$^4$ cm$^2$ mol$^{-1}$ and 7.7 × 10$^4$ cm$^2$ mol$^{-1}$) are the absorption coefficients of α and β phase at the characteristic wavenumber at 766 cm$^{-1}$ and 875 cm$^{-1}$.

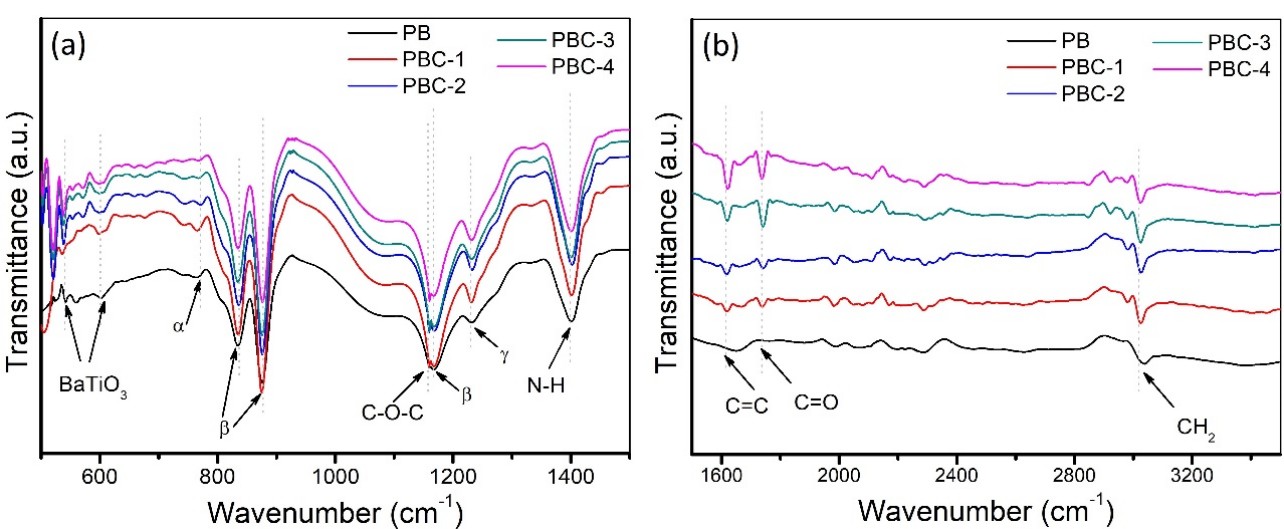

**Figure 5.** FTIR results for the as-prepared PVDF–BaTiO$_3$ (PB) and PVDF–BaTiO3-xCNT (PBC) films within (**a**) 500–1500 cm$^{-1}$ and (**b**) 1500–3500 cm$^{-1}$ wavenumber range.

**Table 1.** Relative proportion of the β-phase in the nanocomposites from FTIR result.

| Nanocomposite | Relative Proportion of β-Phase |
| --- | --- |
| PB | 56–60% |
| PBC-1 | 70–75% |
| PBC-2 | 75–80% |
| PBC-3 | 85–90% |
| PBC-4 | 65–70% |

*3.2. Dielectric Properties*

The dielectric properties for all samples were checked at room temperature (25 °C) over the frequency ranges from 0.1 kHz to 1 MHz after depositing circular Pt electrodes (area: 0.5 mm$^2$) on the back side of the films by RF magnetron sputtering technique. The as-prepared piezoelectric films with ~50 μm thickness and 1 cm$^2$ area were used to investigate the dielectric properties of the samples. Figure 6 depicts the frequency dependence of the dielectric constant and dielectric loss of as-prepared nanocomposites with varying concentrations of MWCNTs. The dielectric constants of all nanocomposites drop as the frequency increases, since the different forms of polarizations cannot keep up with the change of the AC frequency and eventually vanish one by one [44]. The interfacial polarizations at the conductor–insulator interface, also known as the Maxwell–Wangner–Sillar (MWS) polarization, are linked with greater dielectric values at lower frequencies. The micro-capacitance structure model, on the other hand, dominates the dielectric constants at higher frequencies, resulting in stable values [44]. As predicted, the dielectric constant rose as the MWCNT content increased (Figure 6a). The increase in dielectric constant with increased MWCNT contents can be attributed to the formation of many micro-capacitors inside the nanocomposites [44,45]. Because the ceramic fillers have a much higher constant than the polymer matrix, the dielectric constant of the nanocomposites usually increases with higher BaTiO$_3$ content. As a result, the majority of the increase in effective dielectric constant of the nanocomposites comes from the increase in the average field in the polymer matrix. The ceramic filler in the as-prepared nanocomposites, in fact, was kept constant. As a result, it is apparent that filling MWCNT improves the dielectric constant of the nanocomposite considerably. Importantly, two neighboring MWCNT nanofillers can be acted as two specific electrodes within the PVDF matrix to form numerous micro-capacitors that can contribute to capacitance improvement. Notably, the improved capacitance caused by these micro-capacitors is significantly related to the increase in dielectric constant [25]. Furthermore, because MWCNTs behave as a p-type semiconductor with holes as the majority carrier, they can trap electrons from the PVDF matrix and recombine with hole carries to form dipoles. However, because both conducting plates are positively charged, these dipoles are not well aligned, which could be due to the presence of a repulsion force in the center of the micro-capacitor. Furthermore, highly aligned dipoles can be created by applying an external electric field (poling process), which increases the dielectric properties even more. Despite the fact that the dipoles are not highly oriented, the addition of MWCNT conductive nanofillers has the potential to eliminate the need for an external electric field, which requires complex and expensive apparatus.

Figure 6b, on the other hand, shows a small increase in dielectric loss at larger MWCNT concentrations, which might be attributed to leakage current because MWCNT offers a greater number of conductive routes into the PVDF-BaTiO$_3$ matrix [44,46]. When the dielectric constant is taken into account, the PBC-3 sample has a lower dielectric loss than the PBC-4 sample. The findings show that including up to a specific quantity of MWCNT into the PVDF-BaTiO$_3$ matrix not only improves the dielectric constant but also reduces dielectric loss. It is also expected that uniform distribution of composite particles with highly fine surface roughness and homogeneity may further enhance the dielectric constant

with reduced dielectric loss. Moreover, controlling the surface charge density may reduce the dielectric loss with enhanced dielectric constant.

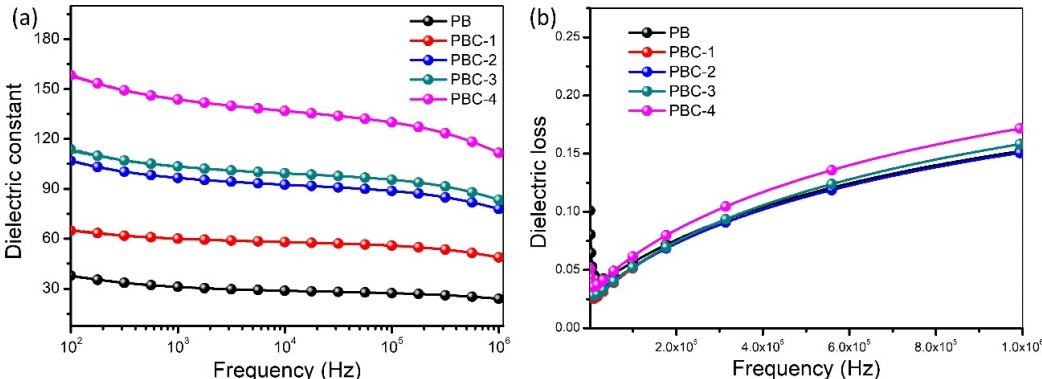

**Figure 6.** Dielectric properties of the as-prepared PB and PBC films—(**a**) dielectric constant and (**b**) dielectric loss.

### 3.3. Energy Harvesting Properties

The ability of the as-prepared samples to generate energy was tested simply by tapping them with a finger (at a force of almost 2 N), and the results are displayed in Figures 7 and 8. The open-circuit output voltage of the samples is shown in Figure 7. A calculated average open-circuit output voltage is summarized in Figure 7f. PBC-3 may generate a maximum of 4.4 V (peak-to-peak), which is approximately three times greater than sample PB, which does not include MWCNT filler. Similar outcome was also observed while testing the current generation capabilities of the samples. Figure 8 shows the peak-to-peak short-circuit output current of the samples. A calculated average short-circuit output current is summarized in Figure 8f. PBC-3 had the highest peak-to-peak output current of 0.66 μA, whereas PBC-2 and PBC-4 had nearly identical results. The results of the as-prepared sample indicated that the integration of MWCNT can minimize the need for further polling of the polymer matrix, improve dipole alignment, and facilitate energy harvesting characteristics. Consequently, it is anticipated that by fine-tuning the BaTiO$_3$ concentrations in the nanocomposites and optimizing the film thickness, the suggested nanocomposite's energy production capabilities may be improved even further.

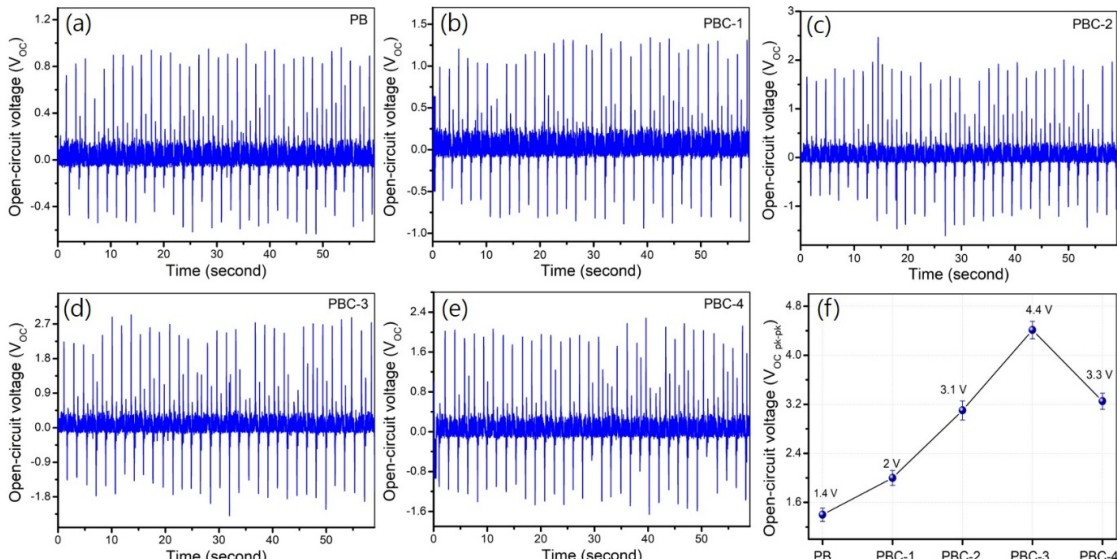

**Figure 7.** Open-circuit voltage of PB and PBC films: (**a**) PB; (**b**) PBC–1; (**c**) PBC–2; (**d**) PBC–3; (**e**) PBC–4; and (**f**) overall peak-to-peak output voltage.

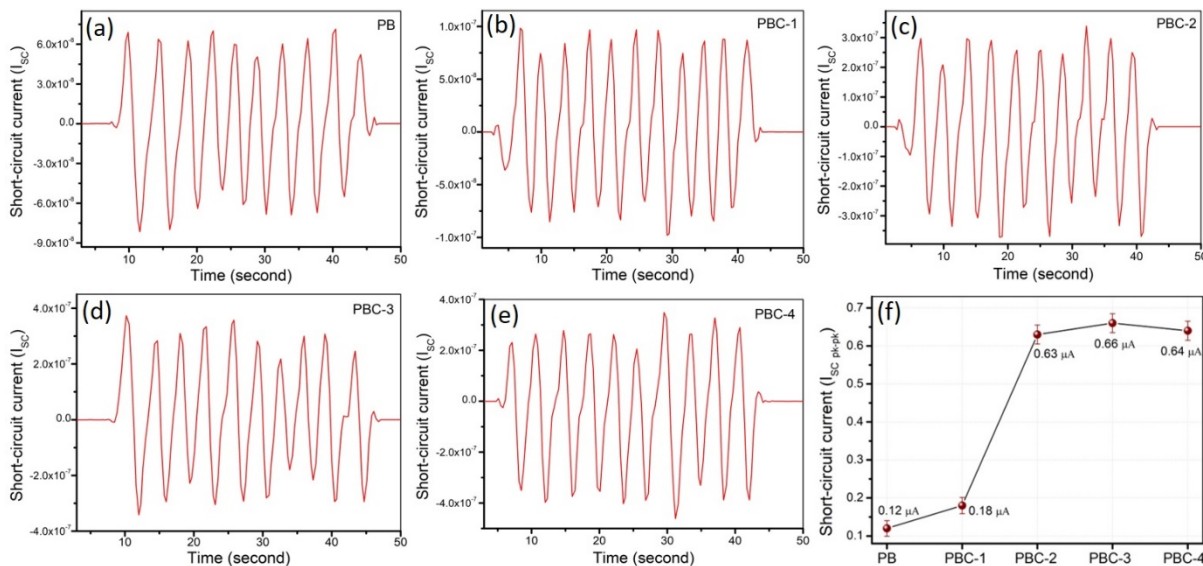

**Figure 8.** Short-circuit current of PB and PBC films: (**a**) PB; (**b**) PBC–1; (**c**) PBC–2; (**d**) PBC–3; (**e**) PBC–4; and (**f**) overall peak-to-peak output current.

## 4. Conclusions

In conclusion, five distinct types of PVDF-BaTiO$_3$ nanocomposites were produced without and with MWCNT nanofiller to study the influence of MWCNT on adjusting dielectric characteristics of polymer-ceramic matrix and their use as piezoelectric nanogenerator (PNG). The results show that including BaTiO3 into the PVDF matrix and then further incorporating MWCNT into the PVDF-BaTiO$_3$ matrix may greatly improve the dielectric characteristics and energy harvesting capabilities of the nanocomposite without the need for any further poling. This illustrates the ease with which high-performance piezoelectric films for energy harvesting may be fabricated. The PBC-3 (PVDF–BaTiO$_3$-0.5 wt% CNT nanocomposite) showed the highest output voltage and current of 4.4 V and 0.66 μA, respectively, at an applied force of ~2 N. It is expected that fine-tuning the BaTiO$_3$ concentration and thickness of the PNGs would improve the harvester's performance even further.

**Author Contributions:** Conceptualization, Investigation, Formal analysis, and Writing—original draft, A.S.M.I.U.; Formal analysis and Validation, D.L.; Investigation and Writing—review & editing, C.C.; Project administration, Supervision, and Visualization, B.K. All authors have read and agreed to the published version of the manuscript.

**Funding:** This research was supported by Basic Science Research Program through the National Research Foundation of Korea (NRF) funded by the Ministry of Education (NRF-2020R1I1A3061814). This research was also supported by Basic Science Research Program through the National Research Foundation of Korea (NRF) funded by the Ministry of Education, Science and Technology (NRF-2020R111A3A04037802). This research was also supported by Basic Science Research Pro-gram through the National Research Foundation of Korea (NRF) funded by the Ministry of Edu-cation, Science and Technology (NRF- -2019R1F1A1062538).

**Institutional Review Board Statement:** Not applicable.

**Informed Consent Statement:** Not applicable.

**Data Availability Statement:** The data created in this study are fully depicted in the article.

**Conflicts of Interest:** The authors declare that there is no conflict of interest.

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
