# Peer review of "Impact of Multi-Walled CNT Incorporation on Dielectric Properties of PVDF-BaTiO3 Nanocomposites and Their Energy Harvesting Possibilities"

_coatings, doi:10.3390/coatings12010077_

Round 1
Reviewer 1 Report
The article is devoted to the study of the properties of nanocomposites based on carbon nanotubes, as well as barium titanate, which has unique ferroelectric properties. Undoubtedly, this line of research is of great interest due to the great prospects for the application of these structures in microelectronics. In the opinion of the reviewer, this article contains new experimental data of novelty and practical significance, which fully correspond to the subject matter of the journal. In general, the article can be accepted for publication after the authors answer a number of the questions posed.
1. The purpose of the work should be reformulated, as well as reflect the prospects for the practical application of these structures.
2. According to the presented SEM images, the distribution of barium titanate in the structure of the composite is uneven, which can lead to inhomogeneity of the dielectric properties of the matrix, can the authors explain this phenomenon, and have there been attempts to achieve a uniform distribution of barium titanate in the structure of the composite?
3. A similar remark applies to the thickness of the coating, what is the reason for its unevenness?
4. The authors, when analyzing X-ray diffraction patterns, mention the presence of two phases in the structure, but do not give their percentage, as well as its changes for different samples.
5. The results of dielectric properties require additional explanations and additions.
Author Response
Response to reviewer comments
Reviewer 1
Thank you very much for your valuable comments and suggestions.
- The purpose of the work should be reformulated, as well as reflect the prospects for the practical application of these structures.
In the current manuscript, we mainly focused to analyze the influence of MWCNT in the dielectric properties of the polymer-ceramic matrix as we can develop a composite for further development of piezoelectric energy harvester.
- According to the presented SEM images, the distribution of barium titanate in the structure of the composite is uneven, which can lead to inhomogeneity of the dielectric properties of the matrix, can the authors explain this phenomenon, and have there been attempts to achieve a uniform distribution of barium titanate in the structure of the composite?
The uneven morphological surface is due to the BaTiO3 particles’ size distribution, which ultimately impact the inhomogeneity and thickness property of the film as well. It is expected that highly uniform particle of BaTiO3 and well distribution will enhance the internal property of the composite. More specifically, spray coating might be a good option. We are trying to make homogenous dispersion of the composite in our future work.
- A similar remark applies to the thickness of the coating, what is the reason for its unevenness?
The uneven morphological surface is due to the BaTiO3 particles’ size distribution, which ultimately impact the inhomogeneity and thickness property of the film as well. It is expected that highly uniform particle of BaTiO3 and well distribution will enhance the internal property of the composite. More specifically, spray coating might be a good option. We are trying to make homogenous dispersion of the composite in our future work.
- The authors, when analyzing X-ray diffraction patterns, mention the presence of two phases in the structure, but do not give their percentage, as well as its changes for different samples.
Due to the uneven distribution of BaTiO3 particles, peak intensities of each sample are not comparable and showed some variations. In this work, we confirmed the presence of the molecules, not the uniformity among the samples as fine tuning is our ongoing work.
The results of dielectric properties require additional explanations and additions.
We suggested that fine tuning of ceramic filler might enhance the composite properties further, which is our future work. Importantly, the current work might not show significant advancement in comparison to other reported works (for example, functionalized polymer or composite with graphene or metal). For this reason, we did not include any comparative analysis with the reported works. Moreover, due to the limitation of required apparatus, we were unable to characterize other related investigations. However, we are trying to develop our work with fining tuning of the composite in terms of ceramic filler and sample thickness. Additionally, we are also focusing energy harvesting application in our future works.
Reviewer 2 Report
This manuscript describes a method to produce five distinct types of PVDF-BaTiO3 nanocomposites w/wo MWCNT nanofillers. The authors examined the influence of MWCNT on adjusting dielectric properties of polymer-ceramic matrix and the application for piezoelectric nano-generator. The incorporation of MWCNT into the PVDF-BaTiO3 matrix greatly improved the dielectric and energy harvesting capabilities. The results shed light on potential applications as mechanoresponsive materials. The experimental results and interpretations are sound. And thus I recommend publishing this manuscript. The specific comments are as follows:
1) What is the surface charge density (μC/m2) and breakdown voltage (V) of the nanocomposites in this study?
2) What is the crystalline content from the XRD study?
3) Page 1, Line 38. polylactic acid (PLLA) perhaps better written as "PLA".
4) I suggest authors to include a table to list the a, β and γ-phase content under different CNT concentrations. This will help reader to better understand the influence of CNT concentration on the β and γ-phase content.
5) I suggest authors to compare d33 values of state-of-the-art and commercial piezoelectric polymers, along with advantages and disadvantages.
6) What are the ways to minimize the dielectric loss shown in Figure 6? I suggest author to include a more in-depth discussion on this part.
Author Response
Response to reviewer comments
Reviewer 2
Thank you very much for your valuable comments and suggestions.
1) What is the surface charge density (μC/m2) and breakdown voltage (V) of the nanocomposites in this study?
Our as-prepared samples contain uneven morphological surface as we have used the BaTiO3 particles with non-uniform size distribution, which ultimately impact the inhomogeneity and thickness property of the film as well. So, we have found varied surface charge density of several samples of the same concentration. For this reason we did not include the result in the manuscript as it is not optimum. Moreover, due to the limitation of required apparatus, we were unable to characterize other related investigations. However, we are trying to develop our work with fining tuning of the composite in terms of ceramic filler and sample thickness.
2) What is the crystalline content from the XRD study?
Due to the uneven distribution of BaTiO3 particles, peak intensities of each sample are not comparable and showed some variations. Moreover, due to the lack of uniformity, we did not include this information in our manuscript. In this work, we confirmed the presence of the molecules, not the uniformity among the samples as fine tuning is our ongoing work. For your acknowledgement, here we attached the raw measurement data:
|
Nanocomposite |
Degree of crystallinity |
|
PB |
38-41% |
|
PBC-1 |
45-48% |
|
PBC-2 |
52-54% |
|
PBC-3 |
37-40% |
|
PBC-4 |
31-34% |
3) Page 1, Line 38. polylactic acid (PLLA) perhaps better written as "PLA".
According to your suggestion, correction is done in the manuscript.
4) I suggest authors to include a table to list the a, β and γ-phase content under different CNT concentrations. This will help reader to better understand the influence of CNT concentration on the β and γ-phase content.
According to your suggestion, calculated relative proportion of the β-phase in the nanocomposites from FTIR result is included in the revised manuscript.
5) I suggest authors to compare d33 values of state-of-the-art and commercial piezoelectric polymers, along with advantages and disadvantages.
The current work might not show significant advancement in comparison to other reported works (for example, functionalized polymer or composite with graphene or metal). For this reason, we did not include any comparative analysis with the reported works. Moreover, due to the limitation of required apparatus, we were unable to characterize other related investigations. However, we are trying to develop our work with fining tuning of the composite in terms of ceramic filler and sample thickness.
6) What are the ways to minimize the dielectric loss shown in Figure 6? I suggest author to include a more in-depth discussion on this part.
According to your suggestion, some information is included in the revised manuscript.
Reviewer 3 Report
1) "Multi-walled carbon nanotubes (MWCNT) are more preferred as dielectric fillers in achieving higher dielectric constant of nanocomposites due to their larger aspect ratio and higher electrical conductivity when compared to spherical and flake shaped fillers."
Explain further and detail about the contribution of high aspect ratio and electrical conductivity to have high dielectric constant. By the way, please explain further about the aspect ratio that author mentioned.
2)"The dielectric properties of the film were measured using an impedance analyzer (KEYSIGHT E4990A)."
What technique and probe were used during measurement?
3)The measured results which associated with dielectric and enegy harvesting properties are key indicator to reflect the attainment of objectives in this work. However, the authors did not relate/discuss closely the dielectric and enegy harvesting results with output of FESEM, EDX and XRD. It seems that these are independent to each other. Result analysis in section 3.3 is not thorough.
Author Response
Response to reviewer comments
Reviewer 3
Thank you very much for your valuable comments and suggestions.
1) "Multi-walled carbon nanotubes (MWCNT) are more preferred as dielectric fillers in achieving higher dielectric constant of nanocomposites due to their larger aspect ratio and higher electrical conductivity when compared to spherical and flake shaped fillers." Explain further and detail about the contribution of high aspect ratio and electrical conductivity to have high dielectric constant. By the way, please explain further about the aspect ratio that author mentioned.
A high aspect ratio indicates long, narrow structure, whereas a low aspect ratio indicates short, wide structure. Aspect ratio can be determined by structure length/structure width. Generally, high aspect ratio can support faster electric conduction as well as recombination within the composite matrix. As in our experiment, we used multiwalled carbon nanotube that has smaller diameter and longer length that is why we mentioned the term high aspect ratio. On the other hand, in comparison to 2D structured filler has lower aspect ratio than 1D structure. For this reason we included the information in our manuscript. Importantly, two neighboring MWCNT nanofillers can be acted as two specific electrodes within the PVDF matrix to form numerous micro-capacitors that can contribute to capacitance improvement. Notably, the improved capacitance caused by these micro-capacitors is significantly related to the increase in dielectric constant.
2)"The dielectric properties of the film were measured using an impedance analyzer (KEYSIGHT E4990A)." What technique and probe were used during measurement?
The system was equipped with a 4-probe work station. The as-prepared samples were measured at room temperature and over the frequency ranges from 0.1 kHz to 1 MHz after depositing circular Pt electrodes (area: 0.5 mm2) on the back side of the films by RF magnetron sputtering technique. Similarly, a continuous thin coating of Pt electrodes (thickness: 120 nm) was deposited on the other side of the film by RF magnetron sputtering technique. The as-prepared films with ~50 mm thickness and 1 cm2 area were used to investigate the dielectric properties of the samples. At various frequency ranges the analyzer acquired the capacitance of the film and recorded. Finally, dielectric constant was calculated using the following formula,
Dielectric constrant, k = (C*d)/e0*A
3) The measured results which associated with dielectric and energy harvesting properties are key indicator to reflect the attainment of objectives in this work. However, the authors did not relate/discuss closely the dielectric and energy harvesting results with output of FESEM, EDX and XRD. It seems that these are independent to each other. Result analysis in section 3.3 is not thorough.
In the current manuscript, we focused the influence of MWCNT in enhancing the dielectric properties of the polymer-ceramic matrix. Moreover, we suggested that fine tuning of ceramic filler might enhance the composite properties further, which is our future work. Morphological and elemental study is closely related to the uniformity, homogeneity and concentration ration of the composite materials. Based on these result, we suggested that fine tuning may enhance our sample’s functionality. On the other hand, XRD and FTIR have also close relation to the formation of highly functional piezoelectric film, which can be explained by the formation of electroactive b phase and we already explained the phenomenon in our manuscript. Moreover, film thickness and surface charge density is also an influencing factor for the enhancement of the dielectric property as well as piezoelectric energy harvester.
As we didn’t get the optimum sample, we didn’t investigate the harvesting properties elaborately. In this work, we just confirmed that the proposed composite has the potential for developing high performance energy harvester. In our ongoing work, we are focusing the harvesting properties with fine tuning of the samples.
Reviewer 4 Report
The submitted manuscript presents fabrication and investigation of the nanocomposites, which contain carbon nanotubes (CNTs), barium titanate (BaTiO3) nanoparticles, and poly(vinylidene fluoride) (PVDF). The morphology, chemical composition, crystal structure, and dielectric properties of these materials were studied. The piezoelectric response of the nanocomposites was examined as a function of the CNT concentration. I think that the topic of the paper and presented results can be interesting for broad range of readers. However, some points have to be clarified or improved. Therefore, I recommend to consider this manuscript for publication after major revision. The following issues should be revised:
- The current status of the piezoelectric nanogenerators should be described in detail in the “Introduction” section. The following literature is recommended and relevant to the topic of the paper: Crystals 11 (2021) 85, Ultrason. Sonochem. 78 (2021) 105718, Sensors 21 (2021) 5144.
- Authors are requested to provide more details on nanocomposites preparation. Were the materials stirred/agitated ultrasonically to obtain homogeneous solutions/suspensions before casting process? What was the temperature of the casted solution/suspension? How about the gas bubbles inside materials? Were they degassed before casting?
- The uncertainty of the films thickness should be determined.
- Were the CNTs-PVDF-BaTiO3 nanocomposites poled before investigation of their piezoelectric properties? What is their coercive filed?
- What was the temperature of the samples during examination of their dielectric and piezoelectric properties?
- What was the sampling frequency applied to register data presented in Figures 7 and 8? Why there are different time constants of the piezoelectric responses given in the Figures 7 and 8?
- How were the values of open-circuit voltage and short-circuit current, shown in Figures 7f and 8f, calculated?
- A comparison table of the piezoelectric performance of the presented CNTs-PVDF-BaTiO3 nanocomposites with other existing piezoelectric nanogenerators can be added to the paper.
Author Response
Response to reviewer comments
Reviewer 4
Thank you very much for your valuable comments and suggestions.
- The current status of the piezoelectric nanogenerators should be described in detail in the “Introduction” section. The following literature is recommended and relevant to the topic of the paper: Crystals 11 (2021) 85, Ultrason. Sonochem. 78 (2021) 105718, Sensors 21 (2021) 5144.
In the current manuscript, we mainly focused to analyze the influence of MWCNT in the dielectric properties of the polymer-ceramic matrix as we can develop a composite for further development of piezoelectric energy harvester. That is why we did not include details on the energy harvester. However, in our ongoing work, we are focusing the harvesting property in details. Moreover, we have included the suggested references in the revised manuscript.
- Authors are requested to provide more details on nanocomposites preparation. Were the materials stirred/agitated ultrasonically to obtain homogeneous solutions/suspensions before casting process? What was the temperature of the casted solution/suspension? How about the gas bubbles inside materials? Were they degassed before casting?
We stirred and ultrasonicated the samples for certain minutes to obtain homogeneous solutions. Before casting we did stirring for 3 hours and sonicated for 5 minutes after every 1 hour of stirring. The solutions were kept in vacuum chamber for 30 min to remove the bubbles from the solution. We included this information in the revised manuscript.
- The uncertainty of the films thickness should be determined.
Unfortunately, we could not determine the uncertainity of the film thickness due to the lake of available equipment.
- Were the CNTs-PVDF-BaTiO3nanocomposites poled before investigation of their piezoelectric properties? What is their coercive filed?
The main objective of the work is to eliminate the process of pooling. So, we did not investigate the samples with pooling process. We got almost similar piezoelectric property of our previous work with applying any external electric field. However, in our ongoing work we are investigating the pooling impact on our samples.
- What was the temperature of the samples during examination of their dielectric and piezoelectric properties?
We maintain the room temperature (25oC) throughtout our experimental process. We mentioned this information in our manuscript.
- What was the sampling frequency applied to register data presented in Figures 7 and 8? Why there are different time constants of the piezoelectric responses given in the Figures 7 and 8?
We tested our sample simply by tapping them with a finger (at a force of almost 2 N). We acquired the outcome within the fixed time range. However, due to the non-uniformity of the tapping frequency, the time constants were slightly non-uniform.
- How were the values of open-circuit voltage and short-circuit current, shown in Figures 7f and 8f, calculated?
The open-circuit voltage and short-circuit current are acquired in a peak-to-peak manner. From the acquired raw data we calculated the average output and presented in the Fig. 7f and 8f. This information is included in our revised manuscript.
- A comparison table of the piezoelectric performance of the presented CNTs-PVDF-BaTiO3nanocomposites with other existing piezoelectric nanogenerators can be added to the paper.
In the current manuscript, we mainly focused to analyze the influence of MWCNT in the dielectric properties of the polymer-ceramic matrix as we can develop a composite for further development of piezoelectric energy harvester. That is why we did not investigate the energy harvesting property in details. As a result, we are unable to include any comparative analysis of the piezoelectric performance of our sample.
Round 2
Reviewer 1 Report
The authors answered all the questions posed. The article can be accepted for publication.
Reviewer 4 Report
The Authors have improved the manuscript taking into serious consideration of the reviewer remarks and suggestions. The revision has been performed. Thus, manuscript can be now accepted for publication.